# Factors Influencing Watching and Purchase Intentions on Live Streaming Platforms: From a 7Ps Marketing Mix Perspective

**Chaang-Iuan Ho [1],*, Yaoyu Liu [2] and Ming-Chih Chen [2]**

[1] Department of Leisure Services Management, Chaoyang University of Technology, Taichung City 413310, Taiwan

[2] Graduate Institute of Business Administration, Fu Jen Catholic University, New Taipei City 242062, Taiwan; lyy19951022@gmail.com (Y.L.); 081438@mail.fju.edu.tw (M.-C.C.)

* Correspondence: ciho@cyut.edu.tw

**Abstract:** Previous studies have investigated how customer purchase intention is influenced by live streaming. However, no study has investigated the effect of service marketing mix (7Ps) on consumer shopping behavior from sellers' perspectives. The present study is designed to shed light on the relationships among the 7Ps and the customers' purchase intention through watching the broadcasters' show. An integrative marketing-oriented model is proposed and tested using data collected from 330 customers (including 237 shoppers for apparel and 93 customers for seafood) through Facebook live shopping platforms. The research results reveal that promotion, placement, and physical evidence have positive effects on customers' purchase intention. In addition, the watching intention had a positive effect on purchase intention. It is also found that watching intention has a full mediation effect on the relationship between the 7Ps marketing mix and the purchase intention. The implications of the findings and issues for future research are also discussed.

**Keywords:** live streaming shopping; marketing mix; purchase intention; watching intention

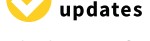



## 1. Introduction

In 2016, Facebook launched its "webcast" feature, attracting users to open live streaming videos that could be watched at any time and anywhere. Many businesses use live webcasts to disseminate business information and sell products, thereby creating another online shopping opportunity. In 2019, the novel coronavirus pandemic shook the world, and due to lockdown and social distancing, consumers were encouraged to use live streaming to shop. Moreover, brick-and-mortar stores suffered from depressed sales due to the impact of the pandemic, which spurred retailers to market their products through live webcasts. Live streaming shopping (LSS) is extremely advantageous because goods can be sold on the Internet, on social media, and in stores, bringing a new sales model to retail e-commerce [1]. One of the attractive features for the audience is the real-time interaction with the broadcaster [2]. The lower selling price than the one in the market serves as a hook that attracts consumers to place orders and enjoy the tension and excitement during the live broadcast [3]. The role of the host is reminiscent of a salesperson at a brick-and-mortar store who introduces the products personally, and this personal touch creates a different feeling from that of other shopping methods [4]. This approach also includes certain unique features such as synchronicity [5] and authenticity [6].

The issue of consumer motivation and purchase intention in LSS has caught researchers' attention [1,3,7–9]. Very little is known from the sellers' perspective [10]. However, business operators have experienced a different level of success with this emerging medium, and the marketing factors that influence consumers' purchase or viewers' followership are still unclear. It also prompts more sellers and marketers to reshape their marketing strategies to better serve their customers.

The features of live streaming make its marketing unique. One of the most important features of LSS is the buying process, which involves a high degree of interdependence between buyers and sellers [10]. Based on a study by Webster [11], in the context of salesperson–consumer interaction, the influencing process involves negotiation rather than persuasion. A long-term personal relationship may be more lasting than product or brand loyalty [12]. In addition, Buttle [13] argued that the product and/or promotion elements may incorporate participants, and the physical evidence and processes may be part of the product. The concept of the 7Ps of the marketing mix [14] may be applied to capture the complexities of the new shopping phenomenon because all service marketing activities are included. The elements of the marketing mix can be controlled by the company and utilized to influence buyer behavior. Ram et al. [15] discussed operational strategies in the context of live streaming commerce by adopting 4Ps. Thus, we believe that the 7Ps concept may be more suitable in studying LSS in a broader context to enhance marketers' competitiveness.

This research attempts to determine the marketing factors that are most important in assessing the willingness to watch live streaming and building shoppers' purchase intention. Thus, we seek to answer the following question: which marketing elements (7P) significantly influence the watching intention and the development of guests' purchasing intention? In spite of the increasing number of people engaging in LS programs/shows, there have so far only been limited explanations of audiences' watching behavior [16,17]. This motivates us to explore viewers' watching behavior on LS platforms by adopting a marketing mix approach that considers customer–seller ties. This study intends to examine the effects of services marketing mix elements on the live streaming commerce (LSC) customers in order to use an appropriate marketing mix strategy, and further explore the mechanism that influences the viewers' watching intentions and related buying behavior.

To achieve the research objective, we develop a marketing-oriented model to empirically investigate the influence of different marketing mix elements in cultivating guests' willingness to watch live streaming and further building their buying intention. By adopting Booms and Bitner's [14] marketing mix theory, we hypothesize that all seven marketing mix elements have a statistically significant impact on shoppers' watching and purchase intentions, as each factor (of the 7Ps) forms an inseparable part of the shopping experience. Previous studies have already demonstrated this theory across a wide range of applications. 7Ps is a collective value proposition offered to customers, and all elements regarding a service or product can be found in the value [18]. The concept has been transformed into quality dimensions to investigate their importance in restaurant customers' loyalty [19] or incorporated into another revolutionary conceptualization, such as Industry 4.0 [20]. Therefore, instead of looking at the relationship between a single element of marketing and live streaming, a more holistic perspective has been employed with the 7Ps framework in the present study. The main contribution of this article is to fill this research gap regarding the determinants of the underlying phenomenon involving the use of LSS by focusing on the seven marketing mix strategies. Due to the rapid growth of live streaming shoppers, a better understanding of the importance and effects of live streaming on marketing may have effective implications for companies and organizations for application in related activities and campaigns.

The remainder of this paper is organized as follows. First, we review the related literature and then develop a conceptual model by identifying constructs from extant literature on the 7Ps of the marketing mix. Next, we present the results of testing the model using empirical data. Finally, we conclude by discussing the implications of these findings for future research and application.

## 2. Literature Review

### 2.1. Live Streaming Shopping and Related Studies

A live broadcast is a form of synchronous social media [5] that includes some unique features, such as synchronicity and authenticity. The former means that all activities of users occur simultaneously [5], and the latter refers to the fact that the video unfolds in an

unedited and somewhat unpredictable fashion, as it allows entrance into the broadcaster's personal life and gives users a sense of reality [6]. Through this approach, users can interact with the show content in real time, allowing instant communication between viewers and live broadcasters [8].

Shopping through live streaming is a new way of making purchases, which not only entails some attributes of e-commerce, but also has distinct characteristics of social media. Social commerce is a way of doing business using social media as a vehicle [21]. Kim and Park [22] defined it as a type of e-commerce that uses social networking sites for social interaction to promote online shopping. LSS has attributes of social commerce, that is, it integrates real-time social interaction with e-commerce [7]. Based on their classifications, LSS can be carried out in the following two ways. (1) Social networking sites are embedded in e-commerce (such as Live.me and Livby). Facebook is currently the most used social networking platform; its original auction community sells through the live broadcast format. (2) E-commerce websites are integrated with social interaction (e.g., Amazon Live Promo Code, Taobao Marketplace and JD.com). A well-known online shopping platform, Shopee, has added a live broadcast function, allowing streamers to sell products during live streaming.

In this study, LSS has been referred to as a form of "livestream + e-commerce," where live streaming is the tool, and e-commerce is the foundation. This operational model overcomes the dual limitations of time and space, allowing consumers to purchase products without leaving their homes and sellers to interact with buyers in real time, thus having an engaging shopping experience and a more interpersonal connection [23]. The LSS platform also provides social activities and entertainment [24]. Live streamers are often content creators who have considerable numbers of viewers; those who consistently watch the broadcast shows are regular followers or potential customers. According to Litovchenko and Shkurupskaya [25], the marketing communication tools in the real space include various types of advertising, sales promotion activities, personal selling, trade shows, and direct marketing. The specific tools in the virtual environment are social media marketing, social media optimization, as well as promotion, forums and virtual exhibitions in a virtual environment. For buyers of LSS, these marketing tools for both the real and virtual spaces can be deployed separately (in parallel) and integrated as a marketing mix.

The emerging phenomenon of LSS has attracted researchers to explore the motivations behind consumers' adoption of this shopping mode and the factors that affect their engagement. The research issues include social learning [26], utilitarian and hedonic motivation [7] and information technology capabilities [1]. Scheibe et al. [5] indicated that the primary motivations of customers include the ease of streaming, the need for self-presentation, boredom and the acceptance of YouNow (a social live broadcast service) by the community. Zhang, Qin, Wang and Luo [27] indicated that live broadcast strategies have an indirect impact on customers' online purchase intention by reducing psychological distance and perceived uncertainty. Wongkitrungrueng and Assarut [23] examined the relationships among the customers' perceived value of live streaming, customer trust and engagement. Chandrruangphen et al. [16] have investigated the effects of LS attributes on customers' intentions to watch and purchase in LSS. Employing the stimulus–organism–response (S-O-R) framework, Xu, Wu and Li [4] investigated contextual and environmental stimuli effects on viewers' cognitive and emotional states and their subsequent responses (hedonic consumption, impulsive consumption, and social sharing) in a live streaming commerce context. In addition, Lee and Chen [3] found that it is easier for consumers in LSS to engage in impulse buying within a short period of time through the presentation and urging of live streamers. Based on the S–O–R theoretical framework, Wang et al. (2021) [28] investigate the impact of live broadcast characteristics on consumers' social presence and flow experience, along with their impact on the consumers' consumption intention. Despite the researchers not adopting the 7Ps marketing mix concept, the attributes were measured as the constructs that were individually found to be associated with personnel, product and price elements. Although the theoretical viewpoints adopted by these studies are

different from that in this research, the analysis has involved some of the elements of the 7Ps marketing mix, providing clues for developing the current research model and the related hypotheses.

### 2.2. Concept of Service Marketing 7Ps

McCarthy [29] proposed the 4Ps of the marketing mix—*product*, *price*, *place* and *promotion*. Later, Booms and Bitner [14] added three elements—*people*, *physical evidence* (the physical surroundings and all tangible cues) and *process*—to propose the new 7Ps of the marketing mix construct. A product is anything that satisfies a want or need and can be offered to a market for attention, acquisition, use, or consumption. Kotler [30] proposed three levels of products—*core*, *actual*, and *augmented products*. The core product is what customers truly want, or it is the substantial benefit customers receive after purchasing the product. The actual product has five characteristics—features, design, quality, packaging, and brand name. The augmented product includes value-added services and benefits. *Place* refers to all enterprises or individuals that acquire or help transfer ownership of a certain type of goods or services. The objective is to eliminate all obstacles in time, location, or ownership between producers and consumers. *Price* refers to how much consumers are willing to pay for a product. Price has a great influence on marketing strategies. There are different pricing strategies according to market positions. *Price* is often used as a competitive tool in the market, including recommended selling prices, cash discounts, and discounts for bulk purchases. *Promotion* is the communication method used by marketers that allows different segments to know the products. In addition to simply informing customers about product-related information, companies can also influence customers' attitudes and motivate them to take action by persuading them. Promotional activities include temporary and monetary or non-monetary incentives. Common promotional tools include price reductions, discount coupons, free samples, gifts, and lucky draws. *People* refers to those who oversee delivery services, that is, service staff. Their clothing, appearance, and attitude affect the quality of service. *Process* refers to the actual procedure and mechanism of delivering services, as well as the role of customers in the operating procedures. A standardized process can improve customers' evaluations of service quality. *Physical evidence* refers to the environment in which the service is delivered, that is, the place where the company and customer interact, and other tangible elements that facilitate communication or the delivery of the service. It can be divided into items such as environmental furnishings, decorations, and service-providing equipment.

### 3. Research Hypotheses Development: Marketing Mix in LSS

Kushwaha and Agrawal (2015) [31] examine the effects of services marketing mix elements on Indian customers in terms of adopting the appropriate marketing mix strategy in the context of banking services. Their rationale is that a marketing mix does not result in a single marketing 'P' strategy, and it may be the interplay of all of the 7P elements at the same time. New marketing channels with the characteristics of interactive media require modifications to their marketing strategies. Information regarding the marketing mix to which consumers are exposed can be regarded as an external stimulus, and external information is the driver of consumer behavior [32]. These marketing strategies may have an impact on consumers' perceptions regarding LSS and purchase intention in relation to the product/service. The present study has reviewed the relevant literature and discovered the 7P features of social commerce. The research hypotheses proposed are as follows. The research framework (Model 1) is presented in Figure 1.

The elements of the marketing mix can be controlled by a company and used to influence buying behavior. New marketing channels with the characteristics of interactive media require modifications in marketing strategies. Information about the marketing mix that consumers are exposed to can be regarded as an external stimulus, and external information is the driver of consumer behavior [32]. These marketing strategies may have an impact on consumers' perceptions about LSS and the purchase intention of the

product/service. The present study has reviewed the relevant literature and discovered the 7P features of social commerce. The research hypotheses proposed are as follows.

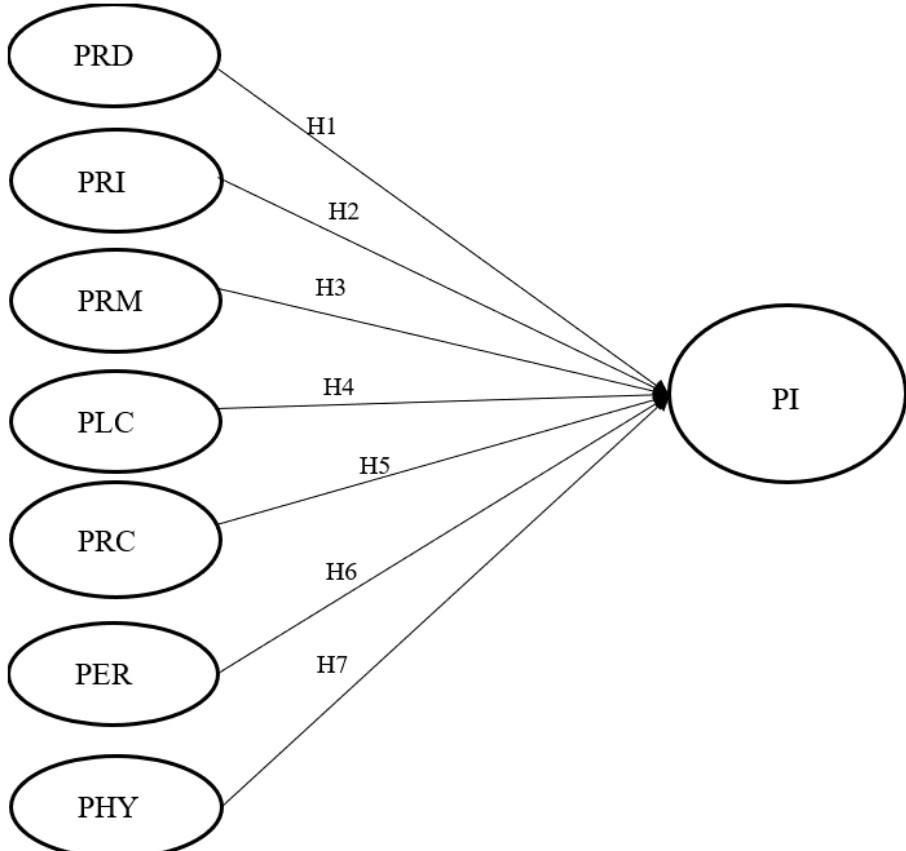

**Figure 1.** Research framework for testing customers' purchase intention (Model 1). Note: Please refer to Appendix A Table A1 for the abbreviations of the constructs.

### 3.1. Product

The key factor for success in online marketing is a product's uniqueness [33], which can easily make the product into the subject of conversation, derive the benefits of word-of-mouth or storytelling marketing, and attract customers' attention. The LSS platform makes it easy for viewers to feel that the products are good value for money and useful [3]. Live streaming commerce has changed e-commerce from the traditional model of "people looking for products" to "products looking for people" [34]. Most products sold through social commerce are general and extremely similar; the advantages of the sellers may be identified if the products sold are unique and differentiated [35]. To attract consumers' attention, products should also be highly functional, practical, and trendy [36]. Thus, the following hypothesis is proposed:

**Hypothesis 1 (H1).** *Products on the LSS platform are positively associated with customers' purchase intention.*

### 3.2. Price

Price is an important factor that affects consumers' purchase intention [33]. Pricing strategies, such as discounts and markdowns, play a critical role in e-commerce [36]. There is less or even no physical store rent in e-commerce, so the selling price is lower. Sellers often propose unique pricing schemes to attract customers and only those who watch the live streams enjoy the discounts and pricing incentives [37,38]. Low prices and

huge discounts are ranked as the main reasons (47.7%) for LSS [39]. Thus, the following hypothesis is proposed:

**Hypothesis 2 (H2).** *The price offered on LSS is positively associated with customers' purchase intention.*

### 3.3. Promotion

The focus of the promotion element is to attract customers by providing a short-term incentive [40]. To arouse customers' curiosity and expectation and then achieve the promotional effect, sellers often announce the start time and content of the broadcast in advance [41]. During the live stream, various interactive games or benefits, such as gift-giving games and lucky draws, are organized to enhance customers' engagement or impulse buying. Such promotions may make sellers more popular and attract new followers [37,42]. Thus, the following hypothesis is proposed:

**Hypothesis 3 (H3).** *The promotions offered on LSS are positively associated with customers' purchase intention.*

### 3.4. Placement

Shopping through a real-time live broadcast is instantaneous and faster than being at brick-and-mortar stores or through textual communication. Consumers are able to interact with other customers who are experts in the products and brands [3]. It allows potential buyers to ask questions or request opinions about the product and interact with salespersons in real time [4,8,42]. The content of live broadcasts is more authentic, and it is easier to gain consumers' trust because it is unedited [27]. Furthermore, a live streaming broadcast provides an entertaining media experience [43]. The live content is more interesting and entertaining because of human interaction, which is a major advantage of LSS [44,45]. Thus, the following hypothesis is proposed:

**Hypothesis 4 (H4).** *The placement (distribution characteristics) of LSS is positively associated with customers' purchase intention.*

### 3.5. People

Attractive endorsers are successful at changing consumers' attitudes and beliefs about a product [46], facilitating purchase decisions and participation behaviors [24]. Live streamers play the role of endorsers of products or brands in LSS [3]. The survey results of the China Consumers Association [47] indicated that customers are mainly attracted by "humorous and funny" (45.9%) and "interesting life" content (44.8%); 30% focus more on the appearance of the streamers. A significant portion of Internet celebrities who match the product image have served as salespersons in live streaming commerce, playing a role in service delivery [4,8]. Their appearance, conversational style and attitude may influence buyers' perceptions about the product/service or cause them to become regular viewers because of the broadcasters' rapport with the audience and personal charisma.

Streamers are integral in the conveyance of the message by professionally providing product information, responding to viewers' questions, making suggestions for purchases and gaining viewers' trust [3,37]. Popularity is also a bonus for attracting customers, especially when well-known Internet celebrities or artists are invited to the platform. With their high popularity and the endorsement for the products, fans have been encouraged to participate in the live broadcast and even add more topics to the show. Viewers may accept the celebrity's recommendation, absorb product information, and adjust their previous perceptions and attitudes accordingly [4]. Thus, the following hypothesis is proposed:

**Hypothesis 5 (H5).** *The personnel of LSS are positively associated with customers' purchase intention.*

### 3.6. Process

Customer experience is highly related to the process of product or service delivery [48]. The service process is related to the service production and delivery. It has been found that the design of the service process indirectly affects the customers' perception [31]. Past research on e-commerce has revealed that factors such as ergonomics or delivery time affect customer satisfaction [49]. Failure to provide fast delivery may prompt consumers to abandon an online shopping platform [36]. Therefore, this research holds that shopping and transaction processes are important elements of live streaming commerce. For example, customers do not need to jump to the webpage to complete the purchase of goods. The shopping model, Message Box +1, facilitates the transaction and brings much convenience to customers. Consumers perceive convenience [34,39,50], including ease of use and saving time, as the main reason for LSS. Thus, the following hypothesis is proposed:

**Hypothesis 6 (H6).** *The process of LSS is positively associated with customers' purchase intention.*

### 3.7. Physical Evidence

Live streaming shoppers often experience the immersive atmosphere of the shopping environment where the decoration, furnishings and configuration make viewers more engaged in the context [27], and this atmosphere easily stimulates their attention and enthusiasm for participation [1]. Information about product appearance becomes more available to customers [4]. Furthermore, customers' experiences are enhanced through various consumption scenarios, such as in-person demonstrations as well as comprehensive and dynamic product displays to spur enthusiasm and elicit consumer purchases [37]. When there is a strong buying interest in the live stream room, viewers are prompted to follow along and place orders. Thus, the following hypothesis is proposed:

**Hypothesis 7 (H7).** *The physical evidence (environment) of LSS is positively associated with customers' purchase intention.*

Viewers' intentions with regard to watching LS programs may play a critical role in generating ongoing revenue. They result from the accumulated online traffic and lead to product/service sales in the future that reflect viewers' behavioral intentions [51,52]. According to a study by Lv et al. (2022) [17], two types of behavior are classified among viewers of tourism LS, namely, immediate buying behavior and continuous watching intention. The former has been regarded as a short-term behavior, and the latter has been denoted as the long-term behavioral intention. The related concepts were adapted from their study and prior research [17,51,52] and modified to fit the context of the present study. Watching intention refers to viewers' ongoing preference for live streams and may be regarded as a proxy for viewers' loyalty, being important for buying behavior [17,51,52]. Faced with increasingly fierce competition in the LSC market, securing and maintaining viewers' watching intentions should be a focus of related enterprises and sellers [52].

According to Babin et al. (1994) [53], customers could be induced to make purchases while they continue to look for more information about the product and receive more information from LSS. In addition, the followership and consumers' purchase intention in LSS are highly positively correlated [54]. We argue that followers may frequently visit live broadcast platforms, send likes and even leave messages (comments). Recently, an empirical study has identified the relationship between consumers' intention to watch a live stream show and their purchase intention [16]. We seek to reveal the importance of the watching intention as a mediating variable. In other words, when the effect of the 7Ps is large, the customers' purchase intention is expected to be even larger with the watching intention as a mediating variable. Thus, another conceptual framework (Figure 2) is presented as follows:

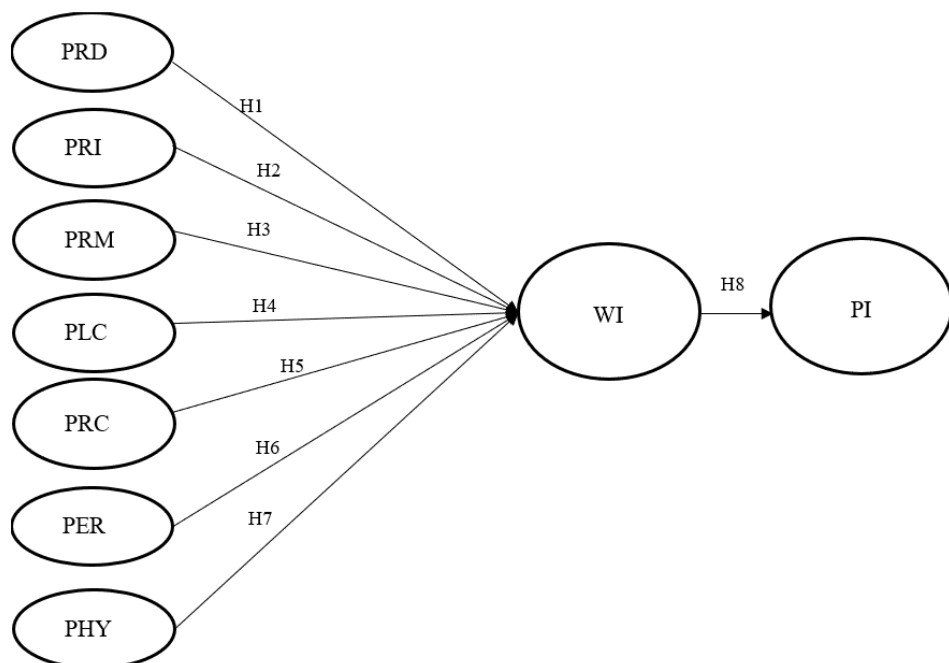

**Figure 2.** Research framework for testing customers' watching intention as the mediator variable (Model 2). Note: Please refer to Appendix A Table A1 for the abbreviations of the constructs.

## 4. Research Methods

Most of the items measuring the chosen research constructs were adapted from prior related research in the field of LSS behavior. Some of the wording for the measurement items was modified to fit the context of this study. This study generated an initial pool of 72 items by reviewing the previous literature and conducting exploratory in-depth interviews. In the interviews, 10 students and 5 adult customers were asked to describe the marketing mix elements that they experienced during an LSS in an open-ended format questionnaire. The purpose of the in-depth interviews was to explore the characteristics not mentioned in previous studies. The interviews were transcribed, analyzed, and converted into items.

Following the development of the original set of statements, the validity of the research instrument was assessed through the face and content validity. The face validity was assessed through three experts who were invited to review the listed items for their appropriateness and completeness. They selected the most precise items for each construct and agreed that the items accurately evaluated the 7Ps concept. After this process, 12 items were eliminated, leaving 60 items and some items were rephrased. Then, content validity was evidenced by conducting the pilot study before starting the formal survey. Thus, a pre-test was undertaken to refine the instrument. The pre-test involved 116 respondents who were asked to evaluate the importance of each measurement item on a five-point Likert scale, ranging from 1 ("extremely unimportant") to 5 ("extremely important"). A criterion was established to eliminate the less important items, that is, those with average scores below 3.5. As a result, 40 items (in Appendix A) were left and formed the basis of the questionnaire in a follow-up survey.

According to the analytics of the LS platforms and live stream auctions [55], apparel and seafood communities on the Facebook live shopping platform attract more customers than other merchant communities. Thus, customers from these two merchant communities were recruited as the sample in the subsequent analysis. The respondents were selected using the convenience sampling method. Through the assistance of live broadcasters (they were given some incentive), we approached viewers or customers to invite them to participate in the survey. The respondents were required to click on an online URL and fill out the questionnaire. Three methods were used to collect data. First, an invitation to

participate in the survey was posted on the community to inform viewers as they watched the live broadcasts. Second, a link to the survey was attached to the receipt of customers. This method prevents the post from being obscured by other information and ensures that each customer receives the survey information. Third, the survey information was placed in the title or body of the live broadcasts; viewers or customers could access the survey link through the high exposure and the verbal promotion of broadcasters. A total of 330 valid responses were obtained, including 237 shoppers for apparel and 93 customers for seafood.

We performed three main statistical analyses to analyze the data collected. We initially employed descriptive statistical analysis to summarize the characteristics of the respondents and the results of the relevant research variables. Regarding the measurement, reference was made to the scale items and to ensure that the referred scale items effectively represented the opinions of the respondents, a preliminary analysis was performed to test the reliability and validity of the scales [56]. A series of exploratory factor analyses (EFA) were conducted to assess the research construct by testing the unidimensionality of each single construct to examine the degree to which the items were tapping the same concept. Then, each scale was subjected to a Cronbach-alpha reliability test, where the values of the alphas were required to be more than the generally accepted required level of 0.7. The detailed estimation results are shown in Appendix B. To assess the potential impact of common method variance, this study also conducted the Harman one-factor test using the confirmatory factor analysis technique [57]. The results indicated that a single-factor model did not fit the data well, suggesting that the common method bias was mitigated.

The proposed model and associated hypotheses were tested using structural equation modeling because both the measurement and the structural model could be evaluated sequentially. The partial least squares structural equation modeling (PLS-SEM) approach was adopted in the present study, and the non-parametric bootstrapping technique was also used. The SmartPLS 2.0 (http://www.smartpls.de/forum/index.php, access on 20 July 2021) software was employed as an analytical tool. Both the measurement model and the structural equation model were tested.

## 5. Research Findings and Discussions

Table 1 provides the demographic information and details of the purchase behavior of the respondents. It reveals that the respondents who bought apparel and seafood through live streaming commerce had different buying habits and socio-economic characteristics. Thus, we compared the responses regarding the 7Ps marketing mix using t-tests. The results indicate that there were significant differences in the two statements about the price, as well as one of the statements about people, process and physical evidence; no discrepancies were found in the other statements. This proves that the items designed by this research reflected the actual situation. In general, the statements represented the marketing mix elements of live streaming commerce.

The assessment of a measurement model should examine (1) reliability, (2) convergent validity and (3) discriminant validity using confirmatory factor analysis (CFA). Table 2 presents the measurement model estimation. The values of composite reliability (CR) for all constructs were above 70, indicating that there was internal consistency. The values of AVE ranged from 0.575 to 0.874, allowing convergent validity. Table 3 provides an overview of the correlation coefficients matrix of the constructs. A comparison of the AVE values with the squared multiple correlations reveals that the AVE values exceed the correlations in all cases, thereby demonstrating discriminant validity for each construct [58]. Overall, the measurement model indicated that there was a high degree of reliability as well as convergent and discriminant validity.

The structural equation model was examined to test the structural equations among the latent constructs, determining their significance and the predictive ability of the model. The bootstrap resampling method (5000 resamples) was used to determine the path coefficients. As presented in Table 4, three hypotheses (H3, H4 and H7) were supported. The effect of promotion on purchase intention was significantly positive (H3). Our results indicate that

place had a positive impact on the purchase intention, supporting H4. In addition, physical evidence (shopping environment) influenced the purchase intention.

**Table 1.** Profiles of respondents (*n* = 330).

| Variable | G1 (%) | G2 (%) | Total (%) | Variable | G1 (%) | G2 (%) | Total (%) |
|---|---|---|---|---|---|---|---|
| **Gender** | | | | **Frequencies of watching live streaming broadcast** | | | |
| Male | 17.2 | 15.2 | 15.8 | Almost everyday | 23.7 | 22.4 | 22.7 |
| Female | 82.8 | 84.8 | 84.2 | 2~4 days per week | 41.9 | 29.5 | 33 |
| **Age** | | | | Once per week | 14 | 8.4 | 10 |
| 20 years old or below | 4.3 | 5.9 | 5.5 | 2~3 days per month | 16.1 | 14.3 | 14.8 |
| 21~30 years old | 16.1 | 35 | 29.7 | Once per month or less | 4.3 | 25.3 | 19.4 |
| 31~40 years old | 31.2 | 44.7 | 40.9 | **Avg. time watching live streaming broadcast** | | | |
| 41~50 years old | 35.5 | 12.7 | 19.1 | 10 min or less | 5.4 | 11.8 | 10 |
| 51~61 years old or above | 12.9 | 1.7 | 4.8 | 11~30 min. | 19.4 | 30.8 | 27.6 |
| **Education** | | | | 31~60 min. | 29 | 21.9 | 23.9 |
| High school diploma | 50.6 | 40 | 43 | 1~2 h | 19.4 | 20.7 | 20.3 |
| Some college | 23.7 | 11.4 | 14.8 | 2~3 h | 15.1 | 10.1 | 11.5 |
| University degree | 23.7 | 42.6 | 37.3 | 3 h or more | 11.8 | 4.6 | 6.7 |
| Graduate school | 2.2 | 5.9 | 4.8 | **Occupation** | | | |
| **Frequencies of live streaming shopping** | | | | Student | 1.1 | 8 | 6.1 |
| Almost everyday | 5.4 | 3.8 | 4.2 | Housewife | 29 | 12.7 | 17.3 |
| 2~4 days per week | 12.9 | 19.4 | 17.6 | Professional services | 8.6 | 8.9 | 8.8 |
| Once per week | 18.3 | 5.9 | 9.4 | Business services | 39.8 | 43.5 | 42.4 |
| 2~3 days per month | 35.5 | 26.2 | 28.8 | Finance and IT | 5.4 | 4.6 | 4.8 |
| Once per month or less | 25.8 | 32.1 | 30.3 | Manufacturing | 7.5 | 12.2 | 10.9 |
| Never | 2.2 | 12.7 | 9.7 | Education and public administration | 5.4 | 5.5 | 5.5 |
| **Marital Status** | | | | Others | 3.3 | 4.6 | 4.2 |
| Married | 62.4 | 42.6 | 48.2 | **Avg. monthly income** | | | |
| Single | 35.5 | 57 | 50.9 | TWD 24,000 or below | 20.4 | 27 | 25.2 |
| Others | 2.2 | 0.4 | 0.9 | TWD 24,001~49,000 | 60.2 | 52.7 | 54.8 |
| **Avg. money amount spent on live streaming shopping** | | | | TWD 49,001~74,000 | 10.8 | 16 | 14.5 |
| Never | 3.2 | 17.3 | 13.3 | TWD 74,001 or above | 8.7 | 4.2 | 5.4 |
| TWD 500 or below | 2.2 | 0.8 | 1.2 | **Live streaming shopping experiences** | | | |
| TWD 501~1000 | 15.1 | 20.3 | 18.8 | Yes | 96.8 | 85.7 | 88.8 |
| TWD 1001~3000 | 58.1 | 42.2 | 46.7 | No | 3.2 | 14.3 | 11.2 |
| TWD 3001~5000 | 12.9 | 13.9 | 13.6 | | | | |
| TWD 5001~7500 | 3.2 | 3.8 | 3.6 | | | | |
| TWD 7501 or above | 5.3 | 17.3 | 2.7 | | | | |

Note: G1 represents the respondents buying seafood, *n* = 93; G2 represents the respondents of buying apparel, *n* = 237.

**Table 2.** Estimation results of measurement model (*n* = 330).

| Constructs/ Variables | Items | Mean | S. D. | Factor Loading | t Value | CR | AVE | Cronbach's α |
|---|---|---|---|---|---|---|---|---|
| PRD | PRD 1 | 4.376 | 0.846 | 0.817 | 7.829 | 0.866 | 0.684 | 0.773 |
| | PRD 3 | 4.730 | 0.591 | 0.853 | 7.251 | | | |
| | PRD 4 | 4.697 | 0.588 | 0.810 | 6.675 | | | |
| PRI | PRI 1 | 4.561 | 0.746 | 0.932 | 6.163 | 0.920 | 0.852 | 0.826 |
| | PRI 2 | 4.706 | 0.610 | 0.914 | 5.293 | | | |
| PRM | PRM 1 | 4.055 | 1.087 | 0.947 | 6.91 | 0.897 | 0.814 | 0.783 |
| | PRM 2 | 4.433 | 0.870 | 0.856 | 5.465 | | | |
| PLC | PLC 1 | 4.591 | 0.710 | 0.771 | 8.238 | 0.899 | 0.598 | 0.868 |
| | PLC 2 | 4.439 | 0.846 | 0.808 | 7.89 | | | |
| | PLC 3 | 4.667 | 0.646 | 0.750 | 8.099 | | | |
| | PLC 4 | 4.658 | 0.648 | 0.797 | 6.798 | | | |
| | PLC 6 | 4.482 | 0.792 | 0.752 | 6.876 | | | |
| | PLC 7 | 4.445 | 0.846 | 0.762 | 6.184 | | | |
| PRC | PRC 1 | 4.679 | 0.614 | 0.819 | 6.163 | 0.841 | 0.639 | 0.717 |
| | PRC 2 | 4.533 | 0.764 | 0.803 | 5.972 | | | |
| | PRC3 | 4.809 | 0.458 | 0.775 | 5.221 | | | |
| PER | PER 1 | 4.715 | 0.549 | 0.858 | 7.758 | 0.906 | 0.708 | 0.863 |
| | PER 2 | 4.585 | 0.666 | 0.876 | 9.235 | | | |
| | PER 3 | 4.667 | 0.577 | 0.866 | 8.378 | | | |
| | PER 4 | 4.809 | 0.458 | 0.763 | 5.009 | | | |
| PHY | PHY 2 | 4.318 | 0.905 | 0.672 | 4.291 | 0.871 | 0.575 | 0.817 |
| | PHY 4 | 4.376 | 0.857 | 0.839 | 8.034 | | | |
| | PHY 5 | 4.642 | 0.643 | 0.737 | 5.683 | | | |
| | PHY 6 | 4.739 | 0.533 | 0.744 | 6.383 | | | |
| | PHY 7 | 4.230 | 0.972 | 0.789 | 8.097 | | | |
| WI | WI 1 | 3.861 | 1.159 | 0.877 | 38.616 | 0.936 | 0.831 | 0.898 |
| | WI 2 | 3.979 | 1.039 | 0.933 | 37.903 | | | |
| | WI 3 | 4.045 | 1.026 | 0.924 | 32.97 | | | |
| PI | PI 1 | 3.597 | 1.220 | 0.901 | 46.466 | 0.954 | 0.874 | 0.927 |
| | PI 2 | 3.594 | 1.127 | 0.952 | 44.57 | | | |
| | PI 3 | 3.524 | 1.178 | 0.950 | 38.35 | | | |

**Table 3.** Descriptive and bivariate correlations between main constructs, and square roots of average variance extracted (AVE).

| | PRD | PRI | PRM | PLC | PRC | PER | PHY | WI | PI |
|---|---|---|---|---|---|---|---|---|---|
| PRD | 0.827 | | | | | | | | |
| PRI | 0.481 | 0.923 | | | | | | | |
| PRM | 0.327 | 0.506 | 0.902 | | | | | | |
| PLC | 0.675 | 0.523 | 0.436 | 0.774 | | | | | |
| PRC | 0.428 | 0.381 | 0.372 | 0.498 | 0.799 | | | | |
| PER | 0.501 | 0.429 | 0.364 | 0.571 | 0.545 | 0.842 | | | |
| PHY | 0.489 | 0.388 | 0.437 | 0.592 | 0.523 | 0.674 | 0.758 | | |
| WI | 0.402 | 0.288 | 0.323 | 0.513 | 0.318 | 0.404 | 0.495 | 0.912 | |
| PI | 0.346 | 0.318 | 0.341 | 0.429 | 0.257 | 0.370 | 0.448 | 0.800 | 0.935 |

Diagonal elements are the square root of average variance extracted. These values should exceed the inter-construct correlations for adequate discriminant validity. Please refer to Appendix A Table A1 for the abbreviationsi of the constructs.

However, our finding was contrary to those of prior research [37], which indicated that service personnel (streamers) are not associated with purchase intention (H5). A possible reason for this difference is that in this study, most respondents may have had LSS experiences and were relatively frequent customers. Although they were familiar with the sellers, experienced consumers were not attracted by the personal characteristics of the broadcaster. Moreover, due to a better understanding of this shopping (business) mode, they were also conversant with information about the product, price and the shopping process. Thus, H1, H2 and H6 were not supported. Clement Addo et al. [54] argued that the

effects of price become insignificant on the customers' purchase intention once they become followers. Lee and Chen [3] excluded product price as the stimulus factor of impulse buying behavior in LSS, which is similar to our finding. However, they found that product usefulness affects customers' perceived usefulness positively. Our finding does not confirm that this factor influenced the consumers' buying intention.

**Table 4.** Hypotheses testing for Model 1.

| Hypotheses | Hypothesized Paths | Coefficients | t-Value | Results |
|---|---|---|---|---|
| H1 | PRD → PI | 0.061 | 0.827 | Not supported |
| H2 | PRI → PI | −0.040 | 0.677 | Not supported |
| H3 | PRM → PI | 0.146 | 2.511 ** | Supported |
| H4 | PLC → PI | 0.170 | 1.972 * | Supported |
| H5 | PRC → PI | −0.059 | 1.099 | Not supported |
| H6 | PEO → PI | 0.042 | 0.608 | Not supported |
| H7 | PHY → PI | 0.254 | 3.982 ** | Supported |

Note: * $p < 0.05$, ** $p < 0.01$.

In order to test the hypotheses regarding the effects of the marketing mix elements on the watching intention, CFA was conducted once again. Figure 3 depicts the model and presents the results of the structural model as well as the standardized path coefficients between the variables.

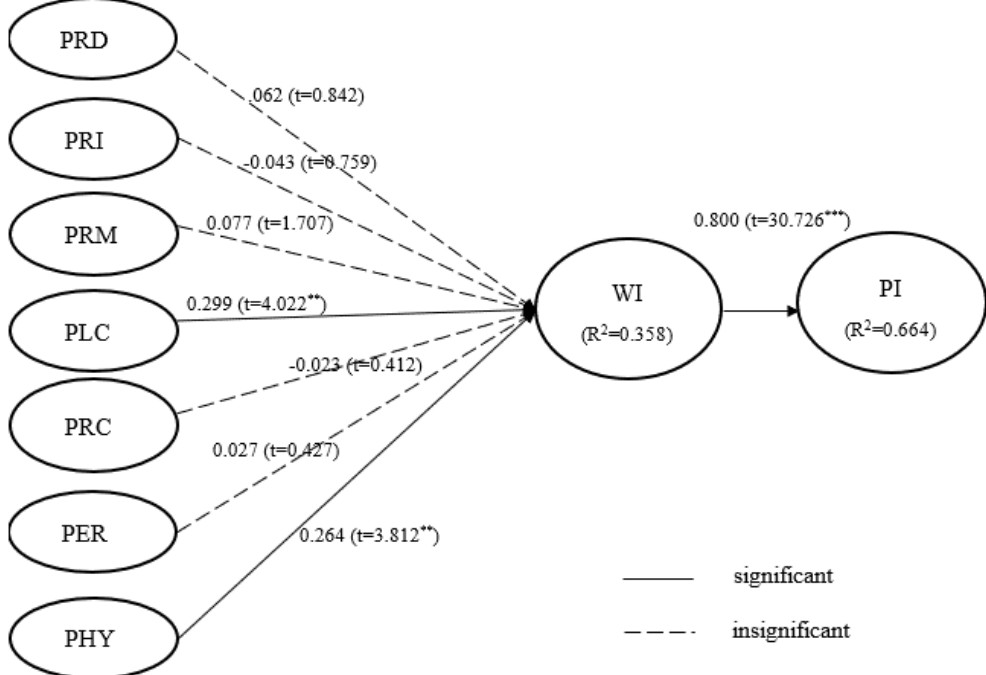

**Figure 3.** Hypothesis testing results for watching intention (Model 2). Note: Please refer to Appendix A Table A1 for the abbreviations of the constructs. ** $p < 0.01$; *** $p < 0.001$.

The position of the service marketing mix as one construct is still ambiguous and not clearly explained. However, the research model of Khatab et al. [59] has dimensions of the service marketing mix as an independent variable and customer satisfaction as a dependent variable. Due to the high degree of correlation among the 7Ps (see Table 3), in referring to [59], we test the mediating effect of the watching intention through the following research framework presented in

The following hypotheses were proposed:

**Hypothesis 8 (H8).** *The service marketing mix is positively associated with customers' purchase intention.*

**Hypothesis 9 (H9).** *The service marketing mix is positively associated with customers' watching intention.*

**Hypothesis 10 (H10).** *Consumers' watching intention is positively associated with their purchase intention.*

**Hypothesis 11 (H11).** *Consumers' watching intention mediates the relationship between the service marketing mix 7Ps and the consumers' purchase intention.*

Table 5 shows that there are two direct relationships and one indirect relationship between variables that have a significant effect. Therefore, H9, H10 and H11 are supported in Model 3. By contrast, there is one direct relationship between the variables that does not significantly influence the relationship between the 7Ps service marketing mix and purchase intention (1.854 < 1.96), thus rejecting H8. In addition, Figure 4 illustrates that the relationship between the watching intention of the 7Ps and purchase intention is one of full mediation. For the reason one can refer to Baron and Kenny [60], who state that if the path coefficient value of indirect influence (c') is greater than the direct effect coefficient (c), then it can be said to have a mediating effect. Furthermore, according to Hair et al. (2010) [61], if a and b are significant, but c is not significant, then the watching intention is expressed as perfect mediation.

**Table 5.** Hypotheses testing for Model 3.

| Relationship between Variables | Effects | Path Coefficient | S.D. | t-Value | Results |
|---|---|---|---|---|---|
| H8: 7Ps → purchase intention | Direct effect | 0.081 | 0.044 | 1.854 | Not supported |
| H9: 7Ps → watching intention | Direct effect | 0.500 | 0.040 | 12.596 | Supported |
| H10: Watching intention → purchase intention | Direct effect | 0.728 | 0.044 | 16.487 | Supported |
| H11: 7Ps → watching intention → purchase intention | Indirect effect | 0.363 | 0.039 | 9.265 | Supported |

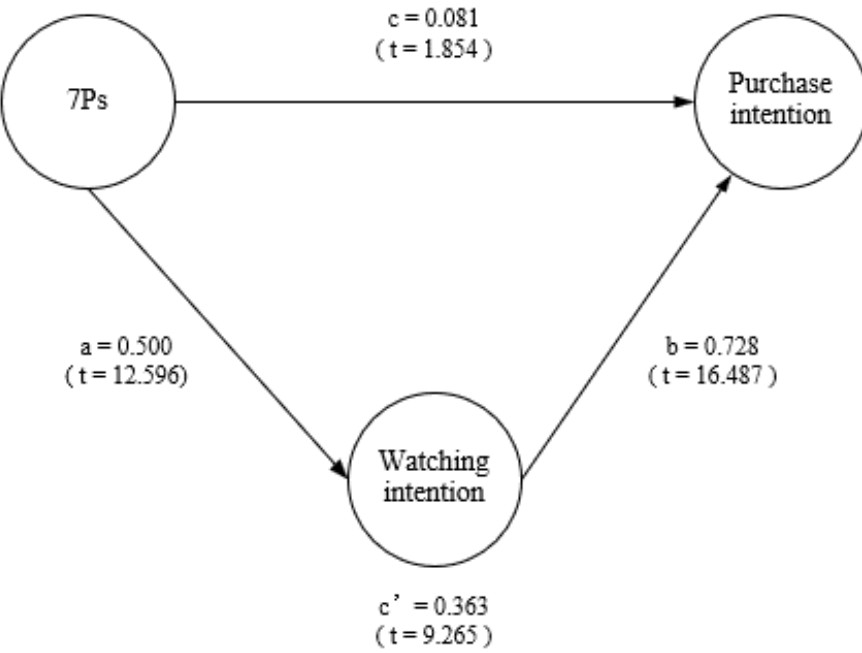

**Figure 4.** The role of watching intention as a mediating variable.

The watching intention is a prerequisite of the purchase intention. Therefore, a live broadcaster (or host) should fully grasp the place characteristics of this marketing channel and create a shopping environment and atmosphere that attract viewers (potential customers) to the communities to watch and enhance the viewers' stickiness. Once viewers have developed the watching habit, they may naturally and gradually make orders. As expected, the shopping process was not found to affect watching intention because the attributes were meaningful to buyers only. It was found that product, price and promotion also did not affect watching intention. A possible reason for this phenomenon is that the respondents may be regular customers who patronize the product. Watching LSS programs may be regarded as a form of entertainment. The most interesting finding was that service personnel (streamers) did not have an effect on the watching intention, which overturns the stereotype that live broadcasters should be good-looking men and women. Perhaps with the passage of time (being used to a certain live broadcaster) and the growing popularity of LSS, the effect of the broadcasters' personal characteristics may be replaced by mutual interactions with customers; service personnel eventually return to the essence of providing professional services. This is because the services of live streaming commerce are superior to those provided by e-commerce.

Our results appear to show some similarities but also some contradictions as compared with Chandrruangphen et al. [16]. As for the similarities, their results indicate that pricing does not have significant influence on consumer intention to purchase, and the intention to watch could lead to intention to purchase. As for the contradiction, their results show that product quality and price transparency significantly influence customer trust and intentions to watch and purchase, while the seller's image of being trustworthy and the quality of the seller's Facebook page only show weak relationships.

## 6. Conclusions

The purpose of this study has been to determine how different marketing elements influence guests' watching and purchase intentions in relation to LSS. Based on the literature review, we have been unable to determine the significance of different marketing elements. Previous studies have discussed some of the elements of the 7P marketing approach and their roles; however, very few studies have analyzed guests' watching and purchasing intentions from a marketing perspective. Thus, we have developed and tested a marketing-oriented model based on a universally comprehensive marketing methodology (7P). The findings are generally verifiable and applicable. By answering the research question posed at the beginning of this study, it is now possible to state that the place characteristics of live streaming (placement) and its physical environment have the greatest impact on guests' watching and purchasing intentions. The present study has also confirmed the findings of previous studies [4,44] and has contributed additional evidence to support the claim that these two elements are the most important marketing strategies in live streaming commerce. Such a novel business model attracts customers to engage in the shopping process due to the dynamic real-time interaction between sellers (streamers) and viewers (buyers). Moreover, it is associated with hedonic factors including an impulse buying atmosphere and environment.

Our findings have also identified the most important indicators of placement and the physical environment. These findings suggest that the promotional strategy can be fulfilled (and/or enhanced) by launching promotional activities. Direct sales are the only form of marketing communication that provides instant feedback and heavily depends on the staff's professionalism. The strategy of physical evidence refers to the environment where a service is performed and delivered with tangible cues that facilitate the delivery of service [62]. It is an important basis for attracting viewers to watch and buy the products/services through the live streaming platform. It is not expected that all the seven marketing strategies will be significant in ensuring viewers' watching intention and their further purchasing intention. This research extends our knowledge of the marketing management of live streaming

commerce. It is the first time that all seven marketing factors (7P) have been examined simultaneously to explore the shoppers' purchasing intention and its implications for LSS.

This research has some limitations. First, the survey was conducted on Facebook Live, excluding other live shopping platforms such as Instagram, and Shopee Live. The research could be replicated across different social media contexts. With advanced technologies and innovative functions, more studies may provide insights into whether the indicators/constructs of the marketing mix are universal. Second, the participants in the study were mainly customers shopping for apparel and seafood. Although the findings of this study have meaningful implications for live streaming commerce, the results might not apply to a wider range of customers and products. The models proposed in this study could be replicated using other groups of respondents (from other regions and with other products) to verify the validity and generalization of the research results. The research results can serve as a measurement tool for assessing the effects of the marketing mix, enabling managers to allocate resources and improve marketing management both precisely and efficiently. Future research could be conducted to compare customers' perceptions regarding marketing-related constructs using a longitudinal assessment. The research results may provide a guideline for assessing the effects of the marketing mix, enabling managers to allocate resources and improve marketing management both precisely and efficiently. Additionally, a further exploration to enhance the conceptualization of the construct of service marketing mix in the LSS context is expected.

It should be noted that female respondents made up the majority of the sample in this study. Although this was common in past studies [1,4], the results should be treated with caution. In future studies, researchers may compare how gender plays a role in customer behavior in live streaming commerce and examine the extent to which a marketing strategy is effective based on gender. Finally, more research on the specific antecedents of the purchase intention of live streaming commerce is needed to explore the strengths of the exploratory factors. The $R^2$ values for both the watching and purchase intentions indicate that there are other influencing variables. Although some marketing mix strategies do not have direct impacts on the watching or purchase intention, they may influence these two consequential variables through other mediators. A more comprehensive model may be an issue for future research.

**Author Contributions:** Conceptualization, C.-I.H.; Data curation, Y.L.; Methodology, C.-I.H.; Writing—original draft, C.-I.H.; Writing—review & editing, M.-C.C. All authors have read and agreed to the published version of the manuscript.

**Funding:** This research was funded by the Ministry of Science and Technology, Taiwan, R.O.C. (No. MOST110-2410-H-324-007).

**Institutional Review Board Statement:** Not applicable.

**Informed Consent Statement:** Informed consent was obtained from all subjects involved in the study.

**Data Availability Statement:** The data that support the findings of this study are available on request from the corresponding author. The data are not publicly available due to privacy or ethical restrictions.

**Conflicts of Interest:** The authors declare no conflict of interest.

## Appendix A

**Table A1.** Measurement Items in the Research Model.

| Constructs/Variables | Items | Source |
|---|---|---|
| Product (PRD) | Products reflect fashion trends (PRD1). Merchandise is sold exclusively on live streaming shopping (PRD2). Products have good prices and are of high quality (PRD3). Products are useful (PRD4). | [62,63] |
| Price (PRI) | Holiday discounts (e.g., the Double 11 Festival) are offered (PRI1). Sellers offer discounts on products (PRI2). Discounted prices are only available during live streaming shopping (PRI3). The price is low than the market price (PRI4). | [64,65] |
| Promotion (PRM) | Lucky draws are often held (PRM1). Some promotional activities (e.g., free gift with purchase, free shipping, buy one get one free, limited time flash sales) are launched (PRM2). The live streaming programs provide the skills for using the products (e.g., teaching how to dress or cook) (PRM3). Sufficient commodity information is provided (such as material, commodity market price, reserve price, and highest bid price) (PRM4). The broadcast time is pre-announced (PRM5). | [42,65] |
| Placement (PLC) | Customers express their opinions by leaving comments at the bottom of the screen (PLC1). I can see the host communicating with customers (PLC2). The host answers customer' questions in real time (PLC3). Customers quickly receive the information delivered by the host (PLC4). The content of the live streaming program is authentic (cannot be edited, pre-recorded, or is difficult to modify/fake) (PLC5) I can see the product reviews of other customers in real time (PLC6). I can watch the show and make an order at any time (PLC7). The live streaming program is funny (PLC8). | [24,44] |
| Process (PRC) | Easy and quick purchase (for example, directly clicking on a link to buy during the live broadcast) (PRC1). There is no need to jump to any interface; customers complete purchase process simply by clicking on the corresponding link (PRC2). There is a smooth network connection and clear pictures during the live broadcast (PRC3). Pays attention to customer privacy and security (PRC4). Diverse payment methods (e.g., credit card, bank transfer, cash on delivery, payment at convenience store) (PRC5). | [51,66] |
| People (PER) | The host is friendly and enthusiastic (PER1). The broadcasting style of the host is interesting (e.g., interesting things to say, having an acting talent) (PER2). The host has good presentation skills to demonstrate products (PEO3). The host has knowledge of the product (PER4). The host is handsome/pretty (PER5). The outfit of the host is in line with the temperament of the product (e.g., fishmongers in frog costumes or mothers selling baby products) (PER6). The hosts are well-known (PER7). | [4,31] |
| Physical Evidence (PHY) | The broadcast room is clean, and the decoration and furnishings are bright and tidy (PHY1). The live broadcast setting matches the style of the products (PHY2). The prices of the products are transparent/visible (PHY3). Customers feel an immersive shopping experience (PHY4). The host personally demonstrates how to use the product (e.g., trials, try-ons) (PHY5). Customers can see the product thoroughly and in detail (PHY6). I can feel the enthusiastic shopping atmosphere (e.g., many shoppers online and stimulating atmosphere of a shopping rush) (PHY7). | [25,67] |
| Watching intention (WI) | If possible, I will continue to watch the broadcasting shows in future (WI1). I plan to watch the shows when I have time (WI2). Even if I don't need to shop, I still plan to watch the shows to gain relevant experience (WI3). | [51] |
| Purchasing Intention (PI) | When I need to buy a particular product, I will consider the way of live streaming shopping (PI1). I plan to shop via live streaming commerce more often in the future (PI2). I prefer live streaming shopping to other shopping approaches (PI3). | [1] |

## Appendix B

**Table A2.** Exploratory Factor Analysis Results of 7Ps.

| 7Ps | Items | Mean | S.D. | Factor Loading | Cronbach's$\alpha$ | Skewness | Kurtosis |
|---|---|---|---|---|---|---|---|
| PRD | PRD 1 | 4.38 | 0.851 | 0.727 | | −1.273 | 1.254 |
| | PRD 3 | 4.74 | 0.582 | 0.871 | 0.719 | −2.518 | 7.293 |
| | PRD 4 | 4.7 | 0.58 | 0.853 | | −2.209 | 6.181 |
| PRI | PRI 1 | 4.57 | 0.742 | 0.847 | | −1.836 | 3.459 |
| | PRI 2 | 4.72 | 0.596 | 0.893 | 0.734 | −2.434 | 7.015 |
| | PRI 4 | 4.69 | 0.62 | 0.687 | | −2.091 | 3.959 |
| PRM | PRM 1 | 4.05 | 1.096 | 0.839 | | −0.923 | 0.040 |
| | PRM 2 | 4.44 | 0.869 | 0.859 | 0.710 | −1.654 | 2.537 |
| | PRM 5 | 4.52 | 0.791 | 0.689 | | −1.839 | 3.561 |
| PLC | PLC 1 | 4.6 | 0.713 | 0.781 | | −2.045 | 4.793 |
| | PLC 2 | 4.44 | 0.854 | 0.774 | | −1.617 | 2.534 |
| | PLC 3 | 4.68 | 0.645 | 0.792 | 0.847 | −2.433 | 7.411 |
| | PLC 4 | 4.66 | 0.65 | 0.822 | | −2.314 | 6.840 |
| | PLC 6 | 4.48 | 0.795 | 0.711 | | −1.705 | 3.320 |
| | PLC 7 | 4.45 | 0.847 | 0.673 | | −1.573 | 2.328 |
| PRC | PRC 1 | 4.68 | 0.615 | 0.714 | | −2.228 | 6.139 |
| | PRC 2 | 4.53 | 0.77 | 0.727 | 0.689 | −1.608 | 2.302 |
| | PRC 3 | 4.81 | 0.462 | 0.788 | | −2.391 | 5.109 |
| | PRC 4 | 4.85 | 0.39 | 0.712 | | −2.606 | 6.387 |
| PER | PER 1 | 4.72 | 0.539 | 0.824 | | −1.934 | 3.491 |
| | PER 2 | 4.59 | 0.664 | 0.848 | 0.835 | −1.465 | 1.352 |
| | PER 3 | 4.66 | 0.581 | 0.852 | | −1.544 | 1.356 |
| | PER 4 | 4.81 | 0.453 | 0.758 | | −2.374 | 5.061 |
| PHY | PHY 1 | 4.6 | 0.699 | 0.693 | | −2.009 | 4.802 |
| | PHY 2 | 4.3 | 0.916 | 0.721 | | −1.155 | 0.669 |
| | PHY 4 | 4.37 | 0.864 | 0.792 | 0.805 | −1.112 | 0.262 |
| | PHY 5 | 4.64 | 0.644 | 0.718 | | −1.729 | 2.224 |
| | PHY 6 | 4.73 | 0.541 | 0.722 | | −2.030 | 3.825 |
| | PHY 7 | 4.22 | 0.98 | 0.689 | | −1.047 | 0.373 |

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
