# Peer review of "Factors Influencing Watching and Purchase Intentions on Live Streaming Platforms: From a 7Ps Marketing Mix Perspective"

_information, doi:10.3390/info13050239_

Round 1

Reviewer 1 Report

This study is an empirical study on Factors influencing purchasing behavior in live streaming from a 7Ps marketing mix perspective. The concept from the perspective of 7Ps marketing mix is meaningful, but I would like to make some important points in the following points.

  1. Abstract lacks impact. It lists only a few formal sentences, and the author(s) would like to write them more organized so that the impact of this study is revealed.
  2. Please present the source of measurement items. If the author(s) developed measurement items themselves, it must go through sufficient reliability and validity verification. The result part only mentioned that it went through CFA, but no verification results were presented.
  3. The bigger problem is, it’s good to think of 7Ps marketing mix concept, but this is only a dimension and not a construct form. According to each dimension, constructs and measurement items used in research should be used. If there is any evidence that the 7Ps marketing mix was directly used in the form of construct, please present it.
  4. In the result part, the watching intention suddenly appeared as a mediation variable. Why didn’t you suggest it as a hypothesis? The focus of this study may vary completely depending on whether or not the WI hypothesis is included. It seems necessary to redesign the story line and structure of the study, starting with the setting of the research problem.
  5. The mediation effect test result has not been presented. It cannot be said that there is a mediating effect without the test results.

Reviewer 2 Report

Congratulaions on your interesting paper.

Everything is clear from the introduction until the conclusions and references.

You have presented a careful literature review and this is important for the theoretical discussion. The sample is quite good for the purpose of the article.

Your discussion section is adequately presented and the tables in research findings and discussions are quite good because they list the main achievements you performed.

Reviewer 3 Report

The paper is concerned with investigating the impact of the 7Ps of marketing mix (product, price, place, promotion, people, physical evidence, and process) on purchasing behaviour in live streaming. The research results are based on the data related to apparel and seafood communities on the Facebook live shopping platform.

In my opinion, the research results have very limited contribution to the theory and practice in the considered field, and consequently, I cannot recommend this paper to publish it in the scientific journal. The authors write “The research results can serve as a measurement tool for assessing the effects of the marketing mix, enabling managers to allocate resources and improve marketing management both precisely and efficiently.” (lines 451-424); however, the research results can be dedicated only to managers (or streamers) who are related to Facebook, and only in two fields, namely apparel and seafood. The title and abstract do not indicate the fact that the research is limited only to data related to apparel and seafood communities on the Facebook. Instead of this, the abstract includes general statements such as “The research results reveal that purchase intention is significantly influenced by three marketing dimensions—(1) promotion, (2) placement, and (3) physical evidence—whereas only the dimension of placement and physical evidence significantly influence the watching intention of broadcasters’ show. The research findings indicate that no other marketing dimension is statistically significant in determining customers’ watching and purchase intentions.” (lines 8-12). In the present form, these statements suggest that the research results refer to purchasing behaviour in all live streaming tools and all fields of shopping.

Round 2

Reviewer 1 Report

Some of the points I made earlier seem to have been supplemented.

However, the following questions still exist:

  1. The logic is still very weak for WI to be included in the research model. Introduction, literature review has not presented the need for the concept of WI and its importance as a parameter, and the rationale and logic of the mediating effect hypothesis are poor. The title doesn't reveal that also.
  2. 7P Marketing mix is the core of this study, and even if several previous studies have been presented, I think logic and theoretical background are still very weak. Looking at individual measurement items, it seems that there is a mixture of conceptual measurements within one dimension. It is desirable to refer to the paper published in the high-quality journal registered in SSCI for previous studies using 7P marketing mix as construction.
  3. Is 7Ps in H2 and H4 2nd order factor? If so, verification process for 2nd order factor should be added. By the way, is there any reason to combine the seven marketing mixes into one single factor? In addition, as shown in Figure 4, the effect of seven marketing mixes on WI individually shows that only PLC and PHY are valid. The other five have no significant impact on the WI, so what's the point of combining these seven into one 2nd order factor to look at the relationship with the WI?
  4. In Line 347, it is mentioned that an exploratory factor analysis was conducted. What tool did you use to go through and how?
  5. In Line 352, Harman’s one-factor test showed that there was no common method bias, and as a basis, “the one-factor model did not fit the data very well” which is a little strange. Please check how to interpret the Harman’s one-factor test results.

Author Response

Dear Reviewer:

Please read the uploaded file.

Reviewer 3 Report

The authors have improved the literature review.

Author Response

(The authors gave the same response as above.)
